# Formulation and Characterization of β-Cyclodextrins–Nitazoxanide Inclusion Complexes: Enhanced Solubility, In Vitro Drug Release, and Antiviral Activity in Vero Cells

**DOI:** 10.3390/pharmaceutics16121494

**Published:** 2024-11-21

**Authors:** Kuppu Sakthi Velu, Sonamuthu Jegatheeswaran, Muhammad Saeed Akhtar, Mohammad Rizwan Khan, Sonaimuthu Mohandoss, Naushad Ahmad

**Affiliations:** 1School of Chemical Engineering, Yeungnam University, Gyeongsan 38541, Republic of Korea; sakthi.velu4@yu.ac.kr; 2BioMe Live Analytical Centre, Karaikudi 630003, Tamil Nadu, India; sjwaran90@gmail.com; 3Department of Chemistry, Yeungnam University, Gyeongsan 38541, Republic of Korea; passions.malik@gmail.com; 4Department of Chemistry, College of Science, King Saud University, Riyadh 11451, Saudi Arabia; mrkhan@ksu.edu.sa

**Keywords:** β-cyclodextrins, nitazoxanide, inclusion complexes, solubility, in vitro drug release, antiviral activity

## Abstract

**Background/Objectives:** Nitazoxanide (NTX) exhibits promising therapeutic potential; its effectiveness is constrained by its low oral bioavailability due to its poor water solubility and limited permeability. **Methods:** This study focused on developing a complex of NTX with β-cyclodextrins (β-CDs), specifically β-CD and hydroxypropyl-β-cyclodextrin (Hβ-CD), to enhance the solubility and antiviral activity of NTX. **Results:** The formation of the β-CD:NTX in an aqueous solution was verified using UV–visible spectroscopy, confirming a 1:1 inclusion complex. Characterization of the solid β-CD:NTX complexes was confirmed via FTIR, X-ray diffraction (XRD), scanning electron microscopy (SEM), and DSC-TGA analyses. Molecular docking studies revealed that the NTX thiazole ring with the nitro group was positioned within the β-CDs cavity, while the benzene ring remained outside. Phase solubility tests showed that β-CD:NTX complexes were formed with high stability constants, demonstrating a linear increase in NTX solubility as the β-CD concentration increased. Dissolution tests revealed rapid and nearly complete NTX release within 90 min for β-CD:NTX and Hβ-CD:NTX complexes. The β-CD:NTX complexes were tested for their antiviral activity against Herpes simplex virus (HSV-1) cultures. Results showed that the Hβ-CD:NTX complex had significantly higher antiviral efficacy than β-CD:NTX and free NTX alone. Moreover, cytotoxicity and cellular uptake studies on Vero cells indicated that the Hβ-CD:NTX complex demonstrated lower cytotoxicity and had the highest IC_50_ value, followed by β-CD:NTX and free NTX. **Conclusions:** These findings suggest that Hβ-CD:NTX inclusion complexes may serve as effective carriers for delivering NTX in HSV-1 treatments using Vero cell models.

## 1. Introduction

Herpes Simplex Virus Type 1 (HSV-1) remains an important global health challenge that affects a large portion of the population and leads to a spectrum of clinical manifestations, which can affect an individual’s quality of life [1]. While many infected individuals remain asymptomatic, those who experience symptomatic outbreaks can suffer from recurrent pain and discomfort, worsening feelings of anxiety, and social isolation because of the visibility of the oral lesions. Furthermore, the potential for HSV-1 to cause severe complications, such as herpes simplex encephalitis, emphasizes the importance of effective management strategies, particularly for vulnerable populations, such as neonates and immunocompromised patients [2]. Despite the effectiveness of current antiviral therapies, including acyclovir and its derivatives, these treatments have considerable limitations [3,4]. Resistance to nucleoside analogs is an emerging concern, particularly in patients with recurrent infections who may require prolonged treatment regimens [5]. This resistance reduces the available therapeutic options and highlights the urgent need for the development of new antiviral agents that can offer an alternative or adjunctive approach to current therapies.

Nitazoxanide (NTX) has emerged as a promising candidate in this context because of its unique mechanism of action and broad-spectrum activity against various pathogens [6]. NTX is classified as a Biopharmaceutics Classification System (BCS) Class IV drug. This means it exhibits low solubility and low permeability. NTX is poorly soluble in aqueous environments, which significantly limits its dissolution and bioavailability. NTX was originally developed to treat gastrointestinal infections caused by protozoa but has demonstrated efficacy against several viral infections, making it an attractive option for further research for antiviral therapy [7]. Its mechanism of action involves the inhibition of pyruvate-ferredoxin oxidoreductase, which is crucial for the energy metabolism of various pathogens, including viruses [8]. The potential of NTX to act against multiple viral targets provides a dual advantage where it effectively inhibits viral replication and may also reduce the likelihood of resistance development, which is an issue with existing antiviral drugs. Furthermore, the ability to explore NTX in combination with established antiviral agents may provide synergistic effects that enhance therapeutic outcomes. Although NTX has considerable therapeutic potential; its clinical application is hindered by its poor water solubility, which substantially limits its bioavailability and thus reduces its therapeutic effectiveness [9]. The crystalline structure and hydrophobic nature of NTX contribute to its low solubility, leading to poor dissolution rates upon oral administration. This presents a critical barrier to achieving the adequate plasma concentration required for effective treatment. As a result, enhancing the solubility of NTX has become a focal point of pharmaceutical research. Various formulation strategies have been explored to improve the solubility and bioavailability of NTX [10]. Recent advances in drug delivery systems, including solid dispersions, lipid-based formulations, and nanotechnology, have shown promise in this regard [11]. Techniques such as spray-drying, freeze-drying, and the incorporation of surfactants have been investigated to enhance dissolution rates, thereby improving its absorption in the gastrointestinal tract [12]. NTX, as the active ingredient, is prepared in a solid form in both suspension and tablet formulations. However, the omission of critical physicochemical properties, such as BCS class (Class IV: low solubility and low permeability), aqueous solubility (reported as very low, approximately 1.65 mg/L in water), pKa (approximately 5.3, affecting its ionization and solubility), and log P (3.9, indicating moderate lipophilicity), leaves gaps in understanding the complete solubility challenges of NTX [13]. Additionally, without details on marketed oral formulations and dosing (commonly available as a 500 mg tablet for parasitic infections), readers lack context for comparing the current findings to existing formulations. Such information would provide a comprehensive baseline for assessing the impact of the cyclodextrin complexes on NTX solubility and therapeutic potential, and its absence limits the scope of the discussion and practical relevance of the findings. Therefore, enhancing the solubility of NTX could provide better therapeutic options for patients.

In recent years, the development of drug formulations that improve the bioavailability of drugs with poor solubility has become an important focus in pharmaceutical research. Poor water solubility remains a common challenge in drug delivery and often limits the therapeutic efficacy of active pharmaceutical ingredients. For many drugs, poor solubility leads to low bioavailability, reduced absorption, and inconsistent drug release profiles, which may negatively impact clinical outcomes [14]. To address these challenges, various strategies have been developed, and among these, cyclodextrins (CDs) and their derivatives have shown particular promise as solubility enhancers [15]. CDs are cyclic oligosaccharides that consist of glucopyranose units, forming a toroidal shape with a hydrophilic exterior and hydrophobic interior. This unique structure allows CD to form inclusion complexes with hydrophobic drug molecules, thereby enhancing their stability and solubility in aqueous environments [16,17]. One of the most widely studied CDs for pharmaceutical applications is hydroxypropyl-β-cyclodextrin (Hβ-CD), a chemically modified β-CD derivative that offers improved aqueous solubility and reduced toxicity. β-CDs are particularly effective in forming inclusion complexes with poorly soluble drugs [18,19,20,21,22,23,24]. Through noncovalent interactions, such as hydrogen bonds, van der Waals (VdW) forces, and hydrophobic interactions, β-CDs can encapsulate drug molecules within their hydrophobic cavity [25]. This encapsulation enhances the solubility, stability, and bioavailability of the drug, allowing for improved therapeutic performance. The inclusion complex is capable of increasing the drugs’ dissolution rate and protecting the active ingredient from degradation, thus extending its shelf life [26]. In the case of NTX, forming complexes with β-CD or Hβ-CD has shown promise in enhancing its solubility profile, leading to increased absorption and improved bioavailability. The oral administration of CDs is considered safe because they are not absorbed through the gastrointestinal tract and therefore do not exhibit toxic effects. These complexes facilitate the solubilization of NTX and also protect the drug from degradation, thus further contributing to its stability and efficacy. The effectiveness of β-CD against *Cryptosporidium parvum* infection has been demonstrated through both in vitro and in vivo studies [27]. Furthermore, Vero cells are often used as a model system to study the efficacy of antiviral compounds, making them an essential component in evaluating the effectiveness of NTX and its β-CD inclusion complexes. Therefore, the enhanced solubility through β-CD inclusion complexes can potentially improve the antiviral activity of NTX in these cells, allowing for more effective targeting of viral infections.

This study focused on enhancing the solubility of NTX through the use of β-CD inclusion complexes and evaluated their antiviral effects specifically against HSV-1, thereby providing a potential new treatment strategy for treating this infection. By investigating the physicochemical properties, stability, and in vitro release profiles of these complexes, this study aimed to improve the therapeutic performance of NTX. The ultimate goal was to develop a more effective formulation that addresses its solubility limitations, thus broadening its clinical application in treating infectious diseases, including viral infections. As the demand for effective antiviral therapies increases, NTX has emerged as a promising candidate for further exploration in the fight against HSV-1.

## 2. Materials and Methods

### 2.1. Materials

NTX was obtained from TCI Chemical (Seoul, Republic of Korea) and was used without further purification. Stock solutions of NTX (1 × 10^−4^ M) were prepared in methanol and stored at 4 °C. β-CD and Hβ-CD were purchased from Sigma-Aldrich and TCI Chemical (Seoul, Republic of Korea). All chemicals used were of analytical reagent grade and were used without additional purification.

### 2.2. Preparation of β-CD:NTX Inclusion Complex

To prepare the β-CDs (β-CD and Hβ-CD) and NTX drug inclusion complex (β-CD:NTX), the co-precipitation method was used, and a 1:1 molar ratio of β-CDs to NTX was established by dissolving 0.307 g of NTX in 20 mL of ethanol and 1 g of β-CDs in 30 mL of water. Both solutions were combined in a sealed glass vial and mixed with a magnetic stirrer, which was stirred at a moderate speed for 30 min at room temperature. Subsequently, the mixture was stirred at a speed of 600 r/min for an additional 2 h to ensure complete interaction. Following this, the final solution was refrigerated at 4 °C for 36 h to promote the formation of the β-CD:NTX inclusion complex. After this incubation period, the precipitated β-CD:NTX inclusion complex was obtained through filtration and thoroughly washed with ethanol to eliminate any uncomplexed NTX drug. The remaining residue was then vacuum-dried for 48 h, thus preparing the β-CD:NTX inclusion complex for further studies.

### 2.3. Characterization Techniques

UV-Vis absorption spectra were measured using an Optizen UV 3220 spectrometer (Busan, Republic of Korea). Solution samples were scanned from 200 to 550 nm at a speed of 240 nm/min, using a quartz cell with a path length of 1.0 cm. FTIR analysis was conducted on β-CD, Hβ-CD, NTX, and their respective inclusion complexes (β-CD:NTX and Hβ-CD:NTX) using KBr pellets and a System 2000 FTIR instrument from Perkin Elmer (Shelton, CT, USA), covering a spectral range from 4000 to 400 cm^−1^. X-ray diffraction (XRD) patterns of β-CD, Hβ-CD, NTX, and their inclusion complexes were recorded using an X-ray diffractometer with CuKα radiation. The instrument settings included a voltage of 30 mA, a current of 30 kV, and a scan rate of 2° min^−1^ within a 2θ angle range of 10–80°. The morphology of β-CD, Hβ-CD, NTX, and their inclusion complexes was examined via scanning electron microscopy (SEM) with a LEO 1430 microscope from Zeiss (Carl Zeiss AG, Jena, Germany). Samples were coated with gold using an Emitech K 550X sputter coater (State College, PA, USA) to improve conductivity before SEM analysis. Thermogravimetric analysis (TGA) was conducted to assess the thermal stability of β-CD, Hβ-CD, NTX, and their inclusion complexes. This analysis was performed with a Shimadzu thermogravimetric analyzer and a Perkin Elmer DSC 7 system. Samples weighing approximately 5–6 mg were heated in alumina crucibles from 50 to 400 °C at a rate of 10 °C/min under a nitrogen atmosphere.

### 2.4. Molecular Docking

To investigate the binding interactions between NTX and β-Cyclodextrins (β-CDs), docking simulations were performed with the help of PatchDock (Beta 1.3 Version) and FireDock (Beta 1.3 Version) servers. Initially, three-dimensional models of NTX and β-CDs (including both β-CD and Hβ-CD) were constructed using Chem3D 21.0.0 software. These models were then saved in PDB format for compatibility with the docking software. Prior to docking, energy minimization was conducted on both the ligand (β-CDs) and the receptor (NTX) to reduce steric clashes and optimize the molecular geometry. The optimized structures were uploaded to the PatchDock server (https://bioinfo3d.cs.tau.ac.il/PatchDock/, accessed on 24 February 2022), which uses a shape complementarity algorithm to predict potential interactions based on surface compatibility. The PatchDock-generated β-CD complexes were subsequently refined on the FireDock server (https://bioinfo3d.cs.tau.ac.il/FireDock/, accessed on 24 February 2022), which evaluates complex stability and binding affinity through an energy-based scoring mechanism. Finally, the top-scoring docked configurations were chosen for further analysis in UCSF Chimera 1.8.1 (https://www.cgl.ucsf.edu/chimera, accessed on 5 April 2022), enabling a detailed examination of structural interactions.

### 2.5. Phase Solubility Studies

To perform phase solubility studies of β-CD and its derivative Hβ-CD with NTX, the procedures previously described by Higuchi and Connors [28] were followed. In this approach, 2 g of NTX is added to 25 mL of aqueous solutions containing various concentrations of β-CD or Hβ-CD (0, 0.002, 0.004, 0.006, 0.008, 0.010, and 0.012 mol/L) in a flask. The mixture was agitated at 30 °C for three days to reach dissolution equilibrium. After this period, the solution was filtered through a 0.45 μm hydrophilic membrane (Fisherbrand™ Membrane Filter, Merck, Seoul, Republic of Korea) filter to remove any undissolved NTX. The concentration of the dissolved NTX in each filtrate was measured at 280 nm via spectrophotometry using a UV-3220 Optizen spectrophotometer. The phase-solubility profile was obtained by plotting the solubility of NTX versus the concentration of β-CD and Hβ-CD. The apparent stability constant (K_S_) was determined from the slope of the linear phase-solubility diagrams.
(1)$${Stability\;constant\;(K}_{s})=\frac{slope}{intercept\times \left(1-slope\right)}$$

### 2.6. In Vitro Drug Release

The in vitro drug release study of the β-CD:NTX and Hβ-CD:NTX inclusion complexes was performed to evaluate the controlled release of NTX under physiological conditions. A precisely weighed sample of the β-CD:NTX and Hβ-CD:NTX inclusion complex containing the equivalent of 25 mg NTX was placed in a 100 mL dissolution vessel with 20 mL of phosphate-buffered saline (PBS) at pH 7.4 and maintained at 37 °C. The dissolution apparatus (708-DS) was set to rotate at a constant speed of 100 rpm to ensure proper mixing and to replicate the gentle stirring environment of bodily fluids. At predetermined time intervals (0.5, 2, 5, 10, 15, 20, 25, 30, 35, 40, 45, 50, 60, 70, 80, and 90 min), 1.0 mL samples of the dissolution medium were carefully withdrawn using a syringe and immediately replaced with an equal volume of fresh PBS to maintain the total volume and prevent fluctuation in the concentration. The withdrawn samples were filtered through a 0.45 µm hydrophilic membrane filter to remove any undissolved particles. The concentration of NTX in the filtered samples was measured using UV–visible spectrophotometry at a wavelength of 422 nm, which is specific for the detection of NTX. The cumulative amount of drug released over time was calculated using a calibration curve prepared from known concentrations of NTX. The release profile of NTX from the β-CD:NTX and Hβ-CD:NTX inclusion complexes was plotted as the cumulative percentage of drug release rate calculated using Equation (2).
(2)$$Drug\;release\;rate\;\left(\%\right)=\frac{The\;amount\;of\;drug\;within\;a\;certain\;period}{Total\;amount\;of\;drug\;entrapped\;in\;\beta -\mathrm{CDs}:\mathrm{NTX}}\times 100$$

### 2.7. Antiviral Activity

To evaluate the antiviral activity of NTX in combination with the β-CDs inclusion complexes, HSV-1 Vero cells were seeded in 96-well plates at a density of 1 × 10^4^ cells/well. The cells were cultured in Dulbecco’s Modified Eagle Medium (DMEM) supplemented with 10% fetal bovine serum (FBS) and incubated at 37 °C in a 5% CO_2_ atmosphere until it reached 80–90% confluence. The β-CD inclusion complexes were dissolved in an antiviral testing medium consisting of DMEM with a reduced FBS concentration. The cells were then infected with HSV-1 at a multiplicity of infection of 0.1 and incubated for 1 h to allow for viral adsorption. After removing the viral inoculum, serial dilutions of the β-CD inclusion complexes were applied to the infected cells. The control groups included untreated infected cells, uninfected cells treated with the β-CDs inclusion complexes, and untreated uninfected cells. The treated cells were incubated at 37 °C in a 5% CO_2_ atmosphere for up to 48 h, depending on the virus and cell line. At 24 and 48 h post-infection, 250 μL of the supernatant was collected and stored at 80 °C until viral titration using the 50% tissue culture infective dose (TCID_50_) method. The results of the viral titration were expressed as log_10_ TCID_50_/mL. Each antiviral assay was performed in triplicate, and the results were presented as the mean ± standard deviation (SD).

### 2.8. Cytotoxicity and Cellular Uptake

The cytotoxicity of free NTX and its β-CD and Hβ-CD inclusion complexes was evaluated using Vero cells. The cells were cultured in DMEM and seeded at a density of 1 × 10^4^ cells/well in 96-well plates. After seeding, the cells were incubated for 24 h at 37 °C in a 5% CO_2_ environment to allow proper adhesion. Various concentrations of free NTX and β-CD inclusion complexes (25, 50, 75, 100, 125, and 150 μg/mL) were prepared in the culture medium, and 100 µL of each concentration was added to the respective wells. Control wells were treated with the culture medium alone or the vehicle control (e.g., dimethyl sulfoxide (DMSO)). The cells were then incubated for 48 h. To assess cell viability, an MTT assay was performed. After incubation for 48 h, 10 µL of MTT reagent (5 mg/mL in PBS) was added to each well, followed by a 4 h incubation. The supernatant was then discarded, and the resulting formazan crystals were dissolved in 100 µL of DMSO. Absorbance was measured at 570 nm using a microplate reader, and cell viability was calculated as a percentage by comparing the absorbance of the treated cells to that of the untreated control cells. All experiments were performed in triplicate. In addition, to evaluate cell morphology, a 4% formaldehyde solution was used to fix the cells, followed by incubation at 25 °C for 1 h. After removing the formaldehyde, the slide chambers were washed once with PBS followed by three additional washes with sterile PBS. Free NTX and its β-CD and Hβ-CD inclusion complexes were prepared at a fixed concentration of 150 μg/mL. For cell staining, 500 μL of 4′,6-diamidino-2-phenylindole (DAPI) and fluorescein isothiocyanate (FITC) were added, and the cells were incubated at 37 °C under a 5% CO_2_ atmosphere. After incubation, the slide chambers were washed three times with sterile PBS, air-dried, and examined under a fluorescence microscope.

## 3. Results and Discussion

### 3.1. UV–Visible Studies of β-CD:NTX Inclusion Complex

The absorption spectra of NTX in a buffer solution at pH 7.4 with varying concentrations of β-CDs (β-CD and Hβ-CD) were measured in the range of 200–550 nm, as shown in Figure 1a,b. The NTX absorption spectrum displayed a strong peak at λ_max_ = 226 nm, along with weaker peaks at λ_max_ = 348 and 422 nm [29]. As the concentration of β-CDs increased (from 0 to 0.012 M), there was a corresponding increase in the absorption intensity of NTX, which suggests an interaction between NTX and β-CDs. Notably, the change in the absorption intensity at the longer wavelength peak (422 nm) was more pronounced than that at the shorter wavelength peak (226 nm), indicating the formation of β-CD:NTX and Hβ-CD:NTX inclusion complexes [22,23]. Furthermore, the characteristic NTX absorption band at 348 nm disappeared in the presence of β-CD:NTX and Hβ-CD:NTX inclusion complexes. A progressive shift of the absorption maxima from the shorter to longer wavelengths was observed, with the peak at 422 nm shifting to 426 nm in the presence of β-CD and to 430 nm in the presence of Hβ-CD. This shift, along with the increase in the absorption intensities, suggests that the solubility of NTX was enhanced because of the hydrophobic interactions between NTX and β-CDs [30].

Moreover, the plot of solubilized NTX versus β-CDs concentration demonstrated a good linear relationship, which suggests the formation of a 1:1 complex, likely because of the inclusion of the thiazole ring within the hydrophobic cavity of the β-CD molecule. The binding constant for the formation of β-CD:NTX and Hβ-CD:NTX inclusion complexes was evaluated by examining the variations in absorbance intensity in relation to the concentration of β-CD. The binding constant (K) was calculated using the modified Benesi–Hildebrand equation [31]. The linearity of the plots of 1/[A-A_0_] versus 1/[β-CD] and 1/[Hβ-CD] confirmed the formation of 1:1 inclusion complexes for both β-CD and Hβ-CD (Figure 1c,d). From the slope and intercept of these plots, the binding constants were determined to be 61.26 M^−1^ for β-CD and 94.18 M^−1^ for Hβ-CD, as shown in Appendix A. The change in the Gibbs free energy (ΔG) for the formation of these inclusion complexes was also calculated using the Benesi–Hildebrand plot and binding constants. The ΔG values were negative for both β-CD:NTX (−10.36 kcal/mol) and Hβ-CD:NTX (−11.45 kcal/mol), indicating that the formation of these inclusion complexes was both spontaneous and exothermic. This suggests that the complexation process is thermodynamically favorable.

### 3.2. Functional Group Analysis of β-CD:NTX Inclusion Complex

The Fourier transform infrared spectroscopy (FTIR) spectra of β-CD, Hβ-CD, NTX, and their respective β-CD:NTX and Hβ-CD:NTX inclusion complexes are presented in Figure 2. The FTIR spectrum of β-CD and Hβ-CD exhibited prominent broad absorption bands at 3500–3000 cm^−1^ (corresponding to O-H stretching vibrations), 3000–2800 cm^−1^ (attributed to C-H and CH_2_ stretching vibrations), 1645–1650 cm^−1^ (H-O-H bending), 1360–1000 cm^−1^ (C-O-C stretching vibrations), and 1155 cm^−1^ (C-O stretching vibrations) [22,23]. In contrast, the FTIR spectrum of NTX displayed characteristic peaks, including an N-H stretching band at 3361 cm^−1^ and an aromatic C-H stretching band at 3089 cm^−1^, along with other significant bands at 1768 cm^−1^ (C=O stretching) and 1532 cm^−1^ (N=O stretching) [32]. The FTIR spectra of both β-CD:NTX and Hβ-CD:NTX inclusion complexes differed from those of pure β-CD and Hβ-CD, particularly in the 1700–1500 cm^−1^ range, where bands associated with the C=O and N=O stretching from NTX were observed. Furthermore, the FTIR spectra of the β-CD and Hβ-CD inclusion complexes showed shifts around the 1656 cm^−1^ region, along with alterations in the intensity and frequency of several other bands. These modifications suggest interactions between the NTX drug and the hydrophobic cavity of β-CD and Hβ-CD, thus confirming the formation of β-CD:NTX and Hβ-CD:NTX inclusion complexes.

### 3.3. Crystalline Properties of β-CD:NTX Inclusion Complex

The X-ray diffraction (XRD) patterns of β-CD, Hβ-CD, NTX, and their β-CD:NTX and Hβ-CD:NTX inclusion complexes are presented in Figure 3. The XRD pattern of β-CD displayed distinct, sharp peaks at 2θ values of 12.3°, 13.6°, 16.8°, 17.5°, 18.7°, 19.3°, 20.4°, 22.1°, 25.6°, 27.5°, and 34.8°, which are characteristic of its crystalline nature [22]. In contrast, the XRD pattern of Hβ-CD was smooth, indicating an amorphous form [23]. The NTX XRD pattern exhibited sharp peaks at 16.4°, 21.1°, 24.7°, 26.3°, 32.5°, 37.8°, and 43.2°, thereby confirming the crystalline nature of NTX [33]. For the β-CD:NTX inclusion complex, the XRD pattern exhibited a significant reduction in both the number and intensity of the diffraction peaks compared to those of pure β-CD and NTX [18,20]. This reduction indicates a disruption of the crystalline structure due to the interaction between β-CD and NTX. The XRD pattern of the Hβ-CD:NTX inclusion complex revealed the complete absence of the characteristic NTX peaks, suggesting that NTX transitioned from a crystalline to an amorphous form [21,22,23], likely because of the encapsulation of NTX within the hydrophobic cavity of Hβ-CD. These changes in the XRD patterns provide compelling evidence for the formation of the inclusion complexes and confirm that the crystallinity of NTX was altered upon complexing with β-CD and Hβ-CD.

### 3.4. Morphology of β-CD:NTX Inclusion Complex

The morphology of β-CDs significantly changed upon forming an inclusion complex with NTX, as shown in Figure 4. These structural alterations are primarily because of a marked reduction in crystallinity, or in some cases, a complete loss of crystalline structure when the guest NTX molecules are incorporated into the β-CDs. The scanning electron microscopy (SEM) images of β-CD, Hβ-CD, NTX, and their β-CD:NTX and Hβ-CD:NTX inclusion complexes are presented in Figure 4. In its native form, β-CD appears as blocky particles of irregular shape and size. This non-uniformity is in contrast with Hβ-CD, which exhibits a spherical morphology with an internal cavity, indicating a different structural arrangement possibly due to the substitution of hydroxypropyl groups [21,22,23]. In contrast, NTX in its free state exhibits a typical crystalline structure, which is consistent with its solid form and reflects its ordered molecular arrangement. Upon forming the inclusion complex with β-CD, the morphology of the β-CD complex becomes multilayered and block-like, indicating a retained crystalline structure; although, it was altered from that of pure NTX or β-CD alone [18]. This suggests that although NTX molecules were embedded within the β-CD structure, some crystallinity was preserved, resulting in a stacked configuration, whereas the Hβ-CD inclusion complex exhibits a marked shift toward an amorphous, irregular structure, likely because of the molecular interactions that disrupt the ordered crystalline form [21]. The hydroxypropyl modification in Hβ-CD may contribute to this shift, as this modified structure potentially allows for more varied molecular orientations when incorporating NTX. These observations collectively indicate evident morphological distinctions between the pure drug, β-CDs, and their respective inclusion complexes, which indicates the successful formation of inclusion complexes. This result is further validated via XRD analysis, which confirmed the transition from crystalline to amorphous forms in the complexes by highlighting the changes in the diffraction patterns.

### 3.5. Thermal Properties of β-CD:NTX Inclusion Complex

The thermal stability of the NTX inclusion complexes with β-CDs was analyzed using thermogravimetric analysis (TGA) and differential scanning calorimetry (DSC) (Figure 5). These techniques provide insight into how the inclusion process affects the NTX thermal properties, as well as the physical and energetic interactions between NTX and the β-CD molecules. TGA was used to compare the thermal decomposition profiles of pure β-CD, Hβ-CD, NTX, and their β-CD:NTX inclusion complexes (Figure 5a). In their pure forms, β-CD decomposed at 275 °C [18], Hβ-CD at 297 °C [19], and NTX at a significantly lower temperature of 195 °C [34], which is consistent with NTX relative thermal instability. The formation of inclusion complexes significantly enhanced the thermal stability of NTX. The β-CD inclusion complex decomposed at 203 °C, followed by additional decomposition at 258 °C and 293 °C, whereas the Hβ-CD complex exhibited decomposition temperatures of 234 °C and 309 °C. This increase in decomposition temperatures for NTX within both the β-CD:NTX and Hβ-CD:NTX inclusion complexes suggests improved thermal stability, likely due to the protective effect of the β-CD hydrophobic cavity, which shields the NTX molecule from heat-induced degradation. The elevated decomposition temperatures observed for the inclusion complexes demonstrate that encapsulation by β-CDs alters the thermal profile of NTX and suggests the successful formation of inclusion complexes [17].

DSC provided further details on the physical and energetic properties of the interactions between NTX and β-CDs (Figure 5b). In the DSC analysis, β-CD exhibited an endothermic peak at 87 °C, which is consistent with the dehydration and release of water molecules from its internal cavity. A second endothermic peak at 325 °C indicated the breakdown of β-CD hydroxyl groups, which is consistent with its melting temperature. For Hβ-CD, similar transitions were observed, with dehydration at 93 °C and melting at 351 °C, thus demonstrating slightly higher thermal stability because of the presence of hydroxypropyl groups. In contrast, pure NTX displayed a distinct melting peak at 210 °C [34], thereby confirming its crystalline nature. Additional minor peaks at 253 °C and 287 °C reflected the characteristic transitions of NTX and emphasized its relatively lower thermal flexibility compared to β-CDs. The DSC curves of the β-CD:NTX inclusion complex showed a shift in the NTX peak to around 203 °C, indicating a change in its thermal behavior. In contrast, the endothermic peak of NTX was completely absent in the Hβ-CD:NTX inclusion complex, suggesting full encapsulation of NTX within the hydrophobic cavity of Hβ-CD. This disappearance also indicates a substantial alteration in NTX crystalline structure, as its melting point was no longer evident in the thermal profile of the inclusion complex. The observed shift and loss of the NTX melting point in the DSC analysis confirm that the NTX molecule was successfully encapsulated within the β-CD structures. This encapsulation modifies its thermal properties and likely contributes to enhanced stability by shielding the NTX within the β-CDs hydrophobic cavities.

### 3.6. Docking Studies of β-CD:NTX Inclusion Complex

Molecular docking studies were performed to model the interaction between NTX and the hydrophobic cavities of β-CDs, including Hβ-CD, to construct β-CD:NTX and Hβ-CD:NTX inclusion complexes. Figure 6 illustrates these models, which were generated using the PatchDock and FireDock servers [35,36]. During docking simulation, various conformations were evaluated, with each conformation scored based on the binding affinity and structural fit. Higher scores indicated a more favorable binding conformation, thus reflecting a more stable and energetically favorable configuration. The PatchDock server provided initial docking conformations by generating β-CD:NTX and Hβ-CD:NTX complexes in a 1:1 binding model, assessing their geometric compatibility, and scoring each based on structural fit [22,33]. The top-ranked models of NTX within β-CDs had the highest scores, which indicated the most favorable binding modes (Appendix A). For the β-CD:NTX complexes, PatchDock assigned geometric complementarity scores of 3664 and 3520, respectively, with interface areas of 441.60 Å^2^ and 437.60 Å^2^. These models revealed strong atomic contact energies of −336.83 kcal/mol for β-CD:NTX and −342.80 kcal/mol for Hβ-CD:NTX, which indicates a robust interaction with NTX embedded within the hydrophobic cavity of β-CD and Hβ-CD.

Further refinement through FireDock enabled more accurate interaction measurements by evaluating the global energy, VdW forces, and atomic contact energies. The global energy values calculated for the β-CD:NTX and Hβ-CD:NTX complexes were −32.16 kcal/mol and −45.23 kcal/mol, respectively. The attractive VdW forces, which facilitate closer molecular contact, were −12.10 kcal/mol for β-CD:NTX and −14.22 kcal/mol for Hβ-CD:NTX, while the repulsive VdW forces were relatively low, at 2.63 kcal/mol and 2.39 kcal/mol. These values indicate a favorable interaction environment within the β-CD and Hβ-CD cavities to stabilize the NTX molecule within the binding pocket. In the optimized docking positions, the NTX molecule was positioned with its thiazole ring fully embedded within the β-CD and Hβ-CD cavities, thus ensuring maximal hydrophobic interactions. In addition, the nitro group of NTX was directed toward the narrower end of the β-CDs cavity, which suggests a specific orientation that minimizes steric hindrance and optimizes the energetic compatibility. This orientation aligns with both the experimental data and predicted stability, thus reinforcing that the NTX molecule was effectively encapsulated within the β-CDs structure. These docking studies revealed that the β-CD:NTX and Hβ-CD:NTX complexes represent highly probable and energetically favorable configurations, as evidenced by the geometric scores, contact energies, and stable VdW interactions [19,22,23]. These results suggest that NTX encapsulation in β-CD and Hβ-CD is structurally and energetically optimized, which corroborates the experimental findings that confirm the formation of a stable inclusion complex. Thus, these docking studies not only support the experimental evidence of NTX binding within β-CD cavities but also provide insight into the spatial arrangement and interaction energies that stabilize these complexes.

### 3.7. Phase Solubility Studies of β-CD:NTX Inclusion Complex

Phase solubility studies are essential for evaluating β-CDs to enhance the solubility of poorly water-soluble drugs, such as NTX. In this study, a phase solubility diagram (Figure 7) was used to examine the NTX solubility profile in the presence of β-CD and Hβ-CD across various concentrations (0.002–0.012 M). The phase solubility plots in Figure 7 show that the apparent solubility of NTX increased linearly with the concentration of both β-CD and Hβ-CD within the tested concentration range. This relationship was well described by the linearity of the plots, with correlation coefficients R^2^ values of 0.9913 for β-CD and 0.9929 for Hβ-CD. The linear relationship, which follows the A_L_-type phase solubility profile as described by Higuchi and Connors [37], is indicative of a 1:1 β-CD:NTX and Hβ-CD:NTX inclusion complex. The slope of the phase solubility plot for both β-CD and Hβ-CD was less than one, which further suggests that the enhanced solubility of NTX can be primarily attributed to the formation of stable, single-molecule inclusion complexes (1:1 stoichiometry) with β-CD and Hβ-CD. These findings indicate that the increased solubility of NTX is due to its encapsulation within the hydrophobic cavity of the β-CDs, thereby reducing its interaction with water and enhancing its apparent solubility [20,21]. To quantitatively assess the stability of these inclusion complexes, the stability constants (K_S_) were calculated using the linear regression of the solubility data. The K_S_ values obtained were 449.1 M^−1^ for the β-CD complex and 731.2 M^−1^ for the Hβ-CD complex, where the higher K_S_ for Hβ-CD suggests a stronger binding affinity and greater stability compared to β-CD. This difference may arise from the modified structure of Hβ-CD, which could provide a better fit or enhanced interaction with NTX, thereby stabilizing the inclusion complex further. The high Ks values for both complexes suggest that the NTX inclusion complexes are highly stable in solution, indicating that β-CDs can significantly improve the aqueous solubility of NTX, which is a promising outcome for pharmaceutical applications.

### 3.8. In Vitro Drug Release of β-CD:NTX Inclusion Complex

The in vitro release profile of NTX from β-CD:NTX and Hβ-CD:NTX inclusion complexes was examined in PBS at pH 7.4. As shown in Figure 8, the release of NTX from these inclusion complexes increased rapidly during the first 50 min and then gradually approached a plateau, indicating a sustained release phase at 90 min. The dissolution rate of NTX from both β-CD:NTX and Hβ-CD:NTX inclusion complexes was significantly higher than that of pure NTX [38]. After 50 min, the release profiles for β-CD:NTX and Hβ-CD:NTX inclusion complexes reached 62.1 ± 1.27% and 67.3 ± 2.01%, respectively, while pure NTX did not reach detectable levels in the solution. By the end of 90 min, the cumulative release reached 84.1 ± 1.81% for β-CD:NTX and 93.6 ± 2.92% for Hβ-CD:NTX, thus surpassing the NTX release rate by more than three times. The improved solubility and release rate of NTX from the inclusion complexes can be attributed to its encapsulation within the hydrophobic cavity of β-CD and Hβ-CD. This encapsulation not only protects NTX but also promotes its gradual release into the aqueous environment. These β-CD complexes increase the drug exposure to the release medium, thereby facilitating faster dissolution [39]. Notably, the Hβ-CD:NTX inclusion complex exhibited a higher release rate than the β-CD:NTX inclusion complex, likely because of the increased hydrophilicity of Hβ-CD, which can enhance water penetration and interaction with the NTX molecule, promoting a faster release. These results demonstrate the effectiveness of the β-CD:NTX and Hβ-CD:NTX inclusion complexes in enhancing the solubility and dissolution rate of NTX.

### 3.9. Antiviral Activity of β-CD:NTX Inclusion Complex

The antiviral effectiveness of β-CD inclusion complexes with NTX was evaluated in vitro to assess whether NTX released from these β-CD:NTX inclusion complexes could inhibit HSV-1 replication in Vero cell cultures [40]. Figure 9a,b present the antiviral efficacy of the β-CD:NTX and Hβ-CD:NTX inclusion complexes over 24 and 48 h intervals. At 24 and 48 h, both β-CD:NTX and Hβ-CD:NTX inclusion complexes significantly reduced the viral titers, closely mirroring the reduction seen with NTX dissolved in ethanol (the positive control). These reductions confirm that both β-CD:NTX and Hβ-CD:NTX inclusion complexes release sufficient NTX to fully inhibit viral replication. The comparable effectiveness of both complexes with the positive control suggests that NTX maintains its antiviral activity against HSV-1 even when delivered via β-CDs complexes [41,42,43]. A notable finding was that the initial reduction in viral titer occurred within the first 6 h. The β-CD:NTX inclusion complex demonstrated a decrease in the viral titer that was nearly equivalent to that of free NTX, indicating the efficient early release of NTX from the complex. However, the Hβ-CD:NTX inclusion complex outperformed both the free NTX and β-CD:NTX inclusion complex in reducing the viral titers at this early stage. This enhanced activity is likely due to the faster release profile of Hβ-CD, which allows NTX to become available to the cells more quickly and effectively. All delivery systems, including the positive control, achieved the maximum reduction in viral titer after 12 h, as shown in Figure 9c,d. This peak antiviral activity suggests that both β-CD:NTX and Hβ-CD:NTX inclusion complexes deliver NTX in a sustained manner that maintains its efficacy and achieves effective viral inhibition within a relatively short period. These findings demonstrate that β-CD:NTX and Hβ-CD:NTX inclusion complexes are promising delivery systems for NTX and improve its antiviral efficiency against HSV-1 infections. The Hβ-CD:NTX inclusion complex, with its enhanced early release, shows particular promise for applications that require fast-acting antiviral action. These results support the potential use of CD-based drug carriers to improve NTX bioavailability and therapeutic effectiveness, potentially allowing for reduced dosing frequency and improved patient compliance. These systems represent an effective approach for developing more efficient and controlled antiviral therapies.

### 3.10. Cell Toxicity and Cellular Uptake of β-CDs:NTX Inclusion Complex

The cytotoxic effects of free NTX and its β-CD:NTX and Hβ-CD:NTX inclusion complexes on normal Vero cells were assessed by exposing the cells to different NTX concentrations (25, 52, 75, 100, 125, and 150 μg/mL) for 48 h. Cytotoxicity was quantified using an MTT assay, and the results are presented in Figure 10a. The MTT assay results demonstrated that the Hβ-CD:NTX inclusion complex exhibited significantly lower cytotoxicity than the free NTX and β-CD:NTX inclusion complex [44]. Cytotoxicity was dose-dependent and increased with higher NTX concentrations. The IC_50_ values at the 150 μg/mL treatment concentration for cell viability were 83.37 ± 2.31% for the β-CD:NTX inclusion complex and 94.09 ± 1.74% for the Hβ-CD:NTX inclusion complex, indicating lower toxicity related to free NTX, which showed an IC_50_ value of 51.08 ± 2.14%. β-CD and Hβ-CD alone, even at 150 μg/mL, exhibited no cytotoxic effects on Vero cells, emphasizing that the β-CD:NTX inclusion complexes are safer for cellular applications than free NTX alone.

A cellular uptake study was performed to evaluate the ability of NTX and its β-CD inclusion complexes to penetrate Vero cells. Fluorescence microscopy with DAPI and FITC stains was used to qualitatively assess NTX uptake. DAPI-stained nuclei (blue fluorescence) and FITC-labeled NTX (green fluorescence) allowed for the visualization of cellular uptake. Figure 10b shows the fluorescence intensities of Vero cells treated with free NTX and β-CD:NTX and Hβ-CD:NTX inclusion complexes, with untreated cells serving as a control. The Hβ-CD:NTX inclusion complex exhibited significantly greater fluorescence intensity than either the free NTX or β-CD:NTX inclusion complex, indicating enhanced cellular uptake [44,45,46]. This suggests that the Hβ-CD:NTX inclusion complex penetrates cells more effectively, potentially because of the improved solubility and transport facilitated by the Hβ-CD-modified structure. This increased cellular uptake of Hβ-CD correlates with the enhanced antiviral activity observed in prior studies, making it a promising delivery system for NTX. These results suggest that the β-CD:NTX and Hβ-CD:NTX inclusion complexes reduce NTX cytotoxicity in Vero cells and improve its cellular uptake, especially with the Hβ-CD:NTX inclusion complex. The reduced cytotoxicity and enhanced uptake indicate that β-CD inclusion complexes, particularly Hβ-CD, offer a promising approach for safer and more effective NTX delivery in antiviral therapies. The combination of low cytotoxicity, high cell viability, and efficient cellular uptake makes these β-CD:NTX inclusion complexes favorable candidates for developing NTX-based treatments for HSV-1 infections.

### 3.11. Mechanism

The enhancement of NTX solubility and bioavailability by β-CD and Hβ-CD is achieved primarily through the formation of inclusion complexes, where NTX is encapsulated within the β-CDs hydrophobic cavity. This encapsulation not only shields NTX from aqueous environments, reducing aggregation and crystallization, but also improves its dispersion, creating an amorphous state with increased surface area for dissolution [47]. The β-CDs act as molecular containers, stabilizing NTX and enhancing its solubility without requiring harsh solvents or additional stabilizers [16]. Hβ-CD, with its modified hydrophilic structure, enhances NTX dissolution further by facilitating faster and more complete release, which can increase NTX drug concentration at absorption sites, improving membrane permeability and cellular uptake [48,49,50,51]. Compared with other solubility enhancement methods like solid dispersions [52], nanoparticles [53,54], and liposomal systems [55], β-CDs offer advantages in simplicity, safety, and stability, especially under physiological conditions [56,57]. Moreover, β-CD and Hβ-CD complexes provide sustained NTX release and improved bioavailability without the physical instability or toxicity risks associated with co-solvents, pH modification, or surfactants. These properties make β-CDs particularly effective for NTX delivery, as they enhance solubility, stability, and bioavailability in a balanced, biocompatible system suitable for antiviral therapies.

## 4. Conclusions

In conclusion, the formulation of β-CDs with NTX demonstrated substantial enhancements in solubility, drug release, and antiviral activity. The UV-visible studies revealed a linear relationship between NTX solubility and β-CDs’ concentration, with binding constants of 61.26 M^−1^ for β-CD and 94.18 M^−1^ for Hβ-CD, which indicates a strong and stable inclusion. Thermodynamic analysis showed negative Gibbs free energy values (ΔG = −10.36 and −11.45 kcal/mol), which suggests the spontaneous and exothermic complex formation of the β-CD:NTX and Hβ-CD:NTX inclusion complexes. The formation of solid β-CD:NTX complexes was characterized via FTIR, XRD, SEM, and DSC-TGA analyses. Furthermore, in silico molecular docking studies supported the experimental results, revealing binding energies of −336.83 kcal/mol for β-CD and −342.80 kcal/mol for Hβ-CD, thus demonstrating strong molecular interactions and confirming the inclusion complexes’ stability. Phase solubility studies revealed a linear increase in NTX solubility with stability constants (Ks) of 449.1 M^−1^ for β-CD and 731.2 M^−1^ for Hβ-CD, indicating stronger complex formation and greater stability with Hβ-CD. In vitro drug release profiles revealed a marked improvement in NTX dissolution, with the Hβ-CD complex achieving 93.6% release at 90 min, compared to 84.1% for β-CD, and significantly surpassing the release of pure NTX. Antiviral activity assays demonstrated that both inclusion complexes effectively inhibited HSV-1 replication, with reduced viral titers comparable to those of NTX dissolved in ethanol, and Hβ-CD exhibited faster and enhanced inhibition at the early stages. Moreover, cytotoxicity assays showed lower cytotoxicity for both inclusion complexes, with IC_50_ values of 83.37 ± 2.31% for β-CD and 94.09 ± 1.74% for Hβ-CD, compared to 51.08 ± 2.14% for free NTX. Furthermore, fluorescence microscopy indicated enhanced cellular uptake of Hβ-CD, which correlates with its improved antiviral activity. These findings highlight the potential of β-CD and Hβ-CD as effective carriers to enhance the solubility, stability, release rate, and antiviral efficacy of NTX, making them promising candidates for improved drug delivery systems in antiviral therapies.

## Figures and Tables

**Figure 1 pharmaceutics-16-01494-f001:**
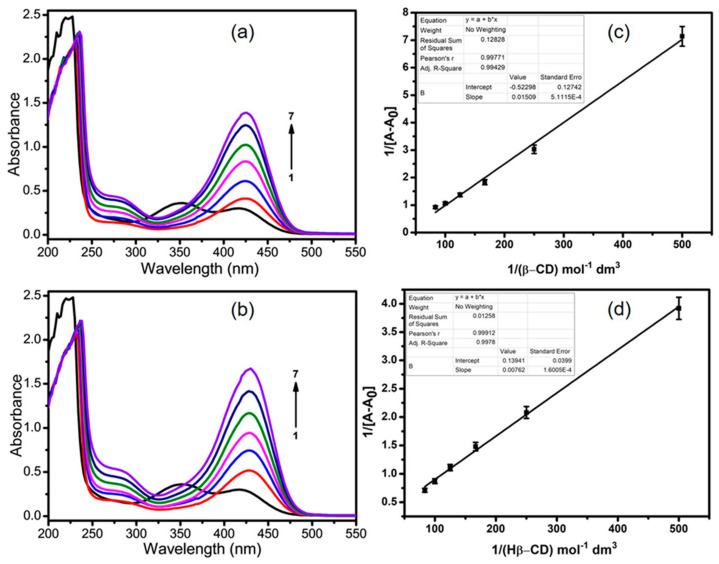
UV–visible absorption spectra of nitazoxanide (NTX) in the presence of (**a**) β-cyclodextrin (β-CD) and (**b**) hydroxypropyl-β-cyclodextrin (Hβ-CD) at pH 7.4 in phosphate-buffered saline solutions. Benesi–Hildebrand plots of 1/[A-A_0_] vs. (**c**) 1/[β-CD] and (**d**) 1/[Hβ-CD]. β-CD and Hβ-CD concentrations ranged from 0 to 0.012 M for measurements 1 to 7 (Black line to Purple line).

**Figure 2 pharmaceutics-16-01494-f002:**
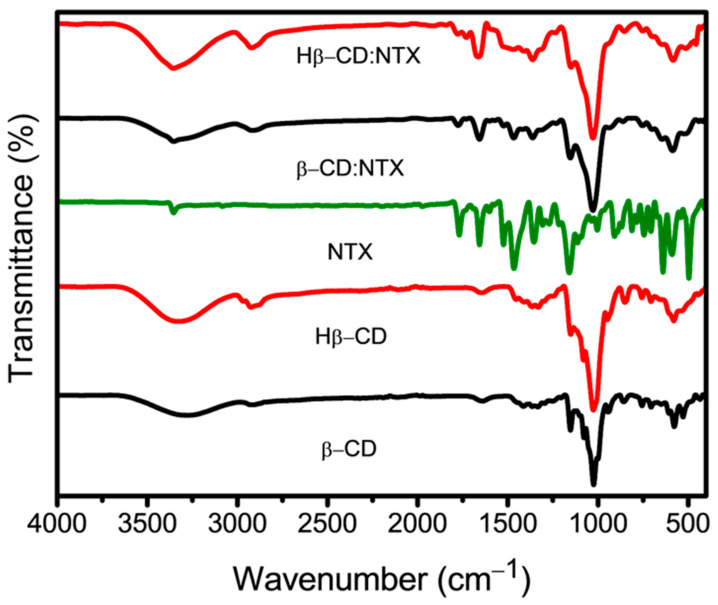
Fourier transform infrared spectroscopy (FTIR) spectra of β-cyclodextrin (β-CD), hydroxypropyl-β-cyclodextrin (Hβ-CD), nitazoxanide (NTX), and their β-CD:NTX and Hβ-CD:NTX inclusion complexes.

**Figure 3 pharmaceutics-16-01494-f003:**
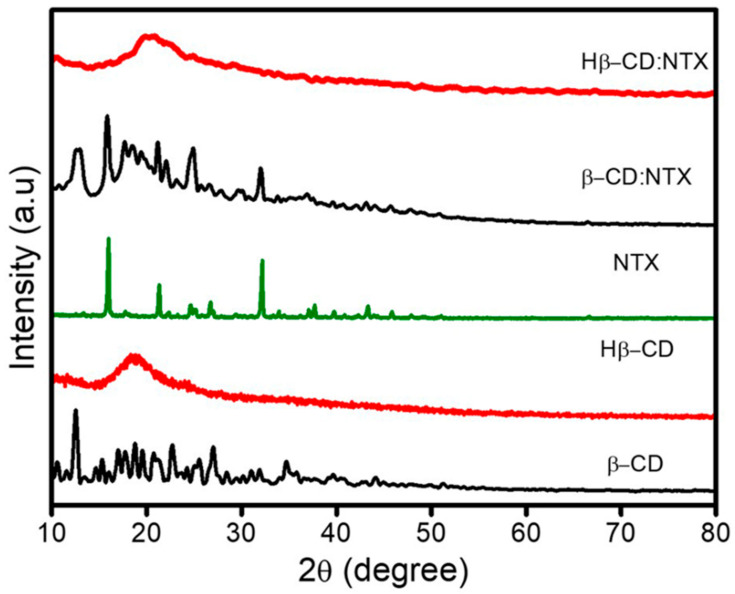
X-ray diffraction (XRD) patterns of β-cyclodextrin (β-CD), hydroxypropyl-β-cyclodextrin (Hβ-CD), nitazoxanide (NTX), and their β-CD:NTX and Hβ-CD:NTX inclusion complexes.

**Figure 4 pharmaceutics-16-01494-f004:**
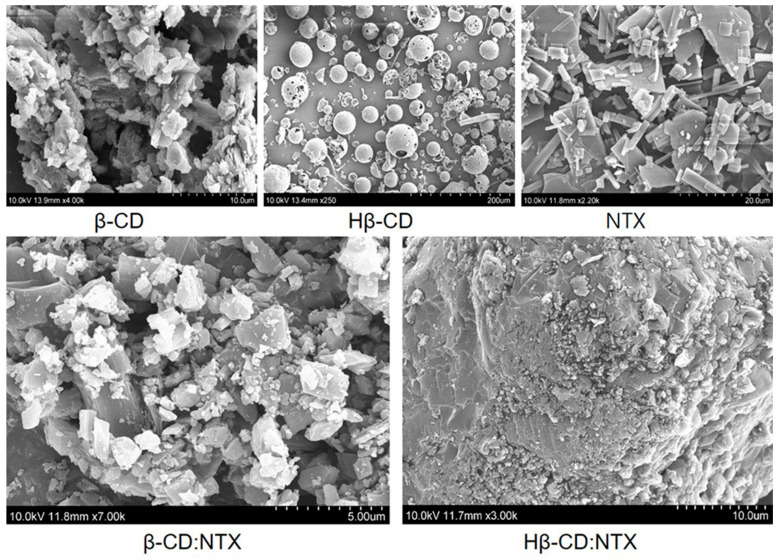
Scanning electron microscopy (SEM) analysis of β-cyclodextrin (β-CD), hydroxypropyl-β-cyclodextrin (Hβ-CD), nitazoxanide (NTX), and their β-CD:NTX and Hβ-CD:NTX inclusion complexes.

**Figure 5 pharmaceutics-16-01494-f005:**
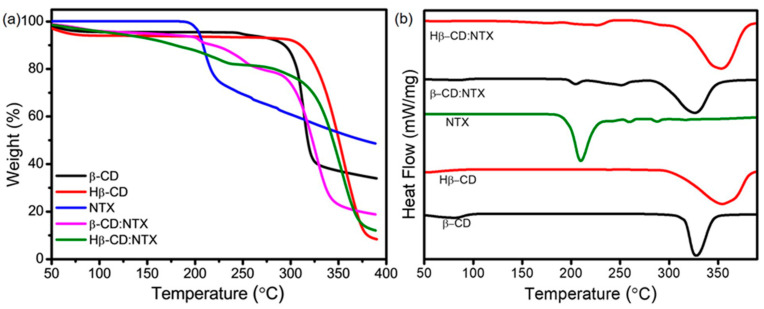
(**a**) Thermogravimetric analysis (TGA) and (**b**) differential scanning calorimetry (DSC) analysis of β-cyclodextrin (β-CD), hydroxypropyl-β-cyclodextrin (Hβ-CD), nitazoxanide (NTX), and their β-CD:NTX and Hβ-CD:NTX inclusion complexes.

**Figure 6 pharmaceutics-16-01494-f006:**
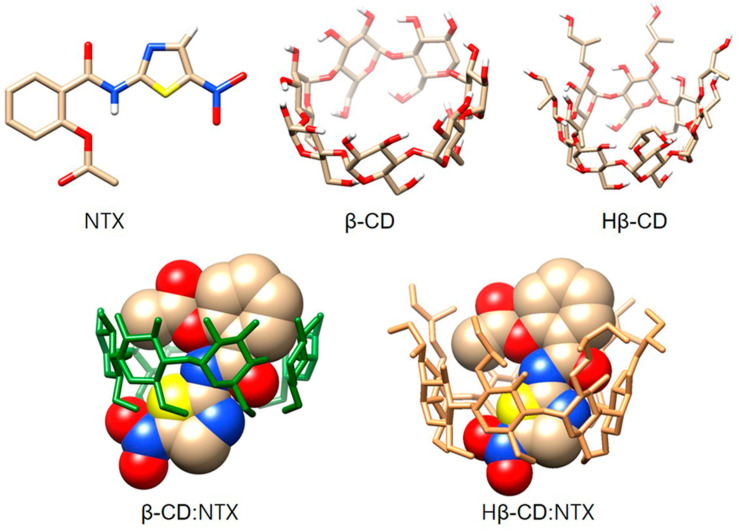
Molecular docking of β-cyclodextrin (β-CD), hydroxypropyl-β-cyclodextrin (Hβ-CD), nitazoxanide (NTX), and their β-CD:NTX and Hβ-CD:NTX inclusion complexes. (NTX; Tan, red, blue, yellow, and white colors corresponding to the carbon, oxygen, nitrogen, sulfur, and hydrogen atoms).

**Figure 7 pharmaceutics-16-01494-f007:**
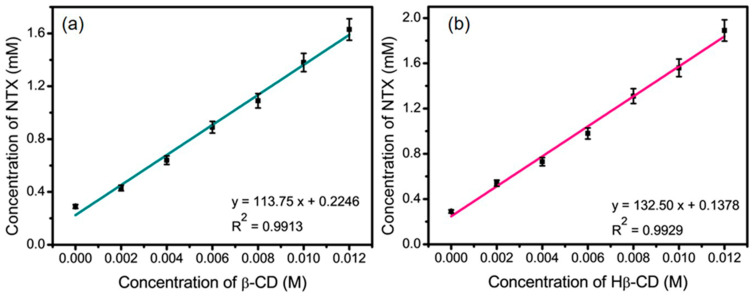
Phase solubility study of nitazoxanide (NTX) in the presence of (**a**) β-cyclodextrin (β-CD) and (**b**) hydroxypropyl-β-cyclodextrin (Hβ-CD).

**Figure 8 pharmaceutics-16-01494-f008:**
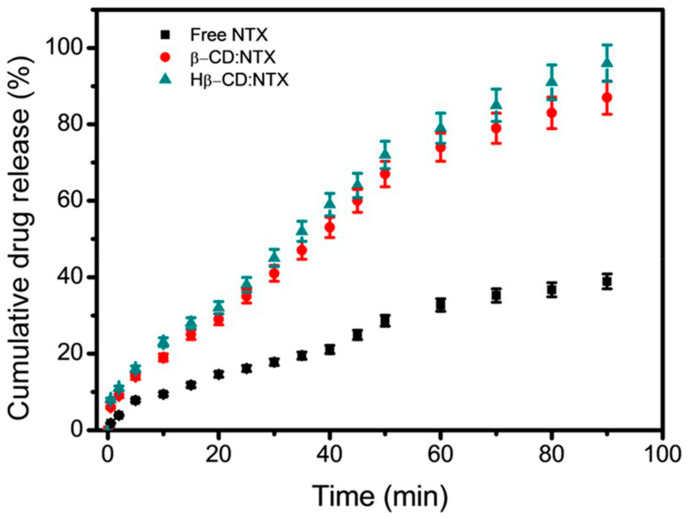
In vitro drug release studies of pure NTX and its β-CD:NTX and Hβ-CD:NTX inclusion complexes.

**Figure 9 pharmaceutics-16-01494-f009:**
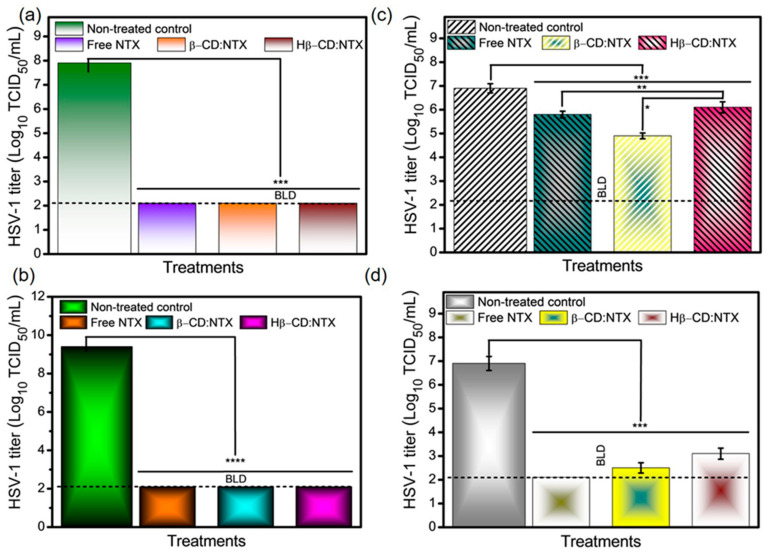
Antiviral activity test at (**a**) 24 h and (**b**) 48 h. Controlled exposure antiviral assay for (**c**) 6 h and (**d**) 12 h (data are the mean ± SD of experiments * *p* < 0.05, ** *p* < 0.01, *** *p* < 0.001, **** *p* < 0.0001), below the limit of detection (using the TCID_50_ method).

**Figure 10 pharmaceutics-16-01494-f010:**
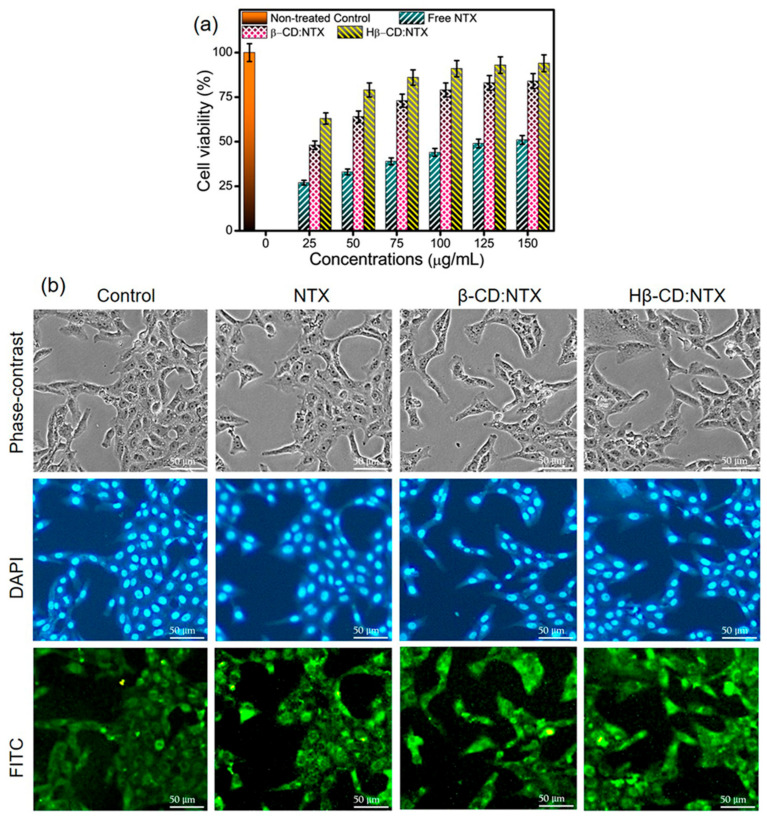
(**a**) Effect of different nitazoxanide (NTX) drug formulations on the viability of Vero cells as a function of free NTX and its β-cyclodextrin (β-CD):NTX and hydroxypropyl-β-cyclodextrin (Hβ-CD):NTX inclusion complexes at 48 h and (**b**) fluorescence microscopy images using phase-contrast, 4′,6-diamidino-2-phenylindole (DAPI), and fluorescein isothiocyanate (FITC) staining of control, NTX, and their β-CD:NTX and Hβ-CD:NTX inclusion complexes at 150 μg/mL for 48 h in Vero cells. Scale bar = 50 μm.

## Data Availability

The original contributions presented in this study are included in this article/Appendix A, further inquiries can be directed to the corresponding author.

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
