# Peer review of "Formulation and Characterization of β-Cyclodextrins–Nitazoxanide Inclusion Complexes: Enhanced Solubility, In Vitro Drug Release, and Antiviral Activity in Vero Cells"

_pharmaceutics, 2024, doi:10.3390/pharmaceutics16121494_

Round 1
Reviewer 1 Report
Comments and Suggestions for Authors
The current research article focusing on development of cyclodextrin inclusion complexes of nitazoxanide for improving oral bioavailability and anti-viral activity is interesting and matches the scope of the journal. However, the current version of the manuscript cannot be accepted for publication. The manuscript needs to be revised in various sections of the methodologies for better understanding. The authors are suggested to address the below comments:
1. What is the BCS class of the drug?
2. Line 36: Be specific which type of cyclodextrin complex is effective.
3. Authors have focused much on discussing the poor solubility of Nitazoxanide but have not provided any information such as BCS class, aqueous solubility, pKa, log P, any marketed oral formulations and doses available.
4. Do authors think co-evaporation is the appropriate term to describe the process of preparing drug and cyclodextrin complexes as described in section 2.2
5. Section 2.3: what type of filter was used? Is it hydrophilic or hydrophobic?
6. Section 2.3: At what wavelength were the samples measured? Also please provide additional information pertaining to the calibration curve.
7. Section 2.3: Cite the reference got Higuchi and Connors method.
8. Section 2.4: What’s the volume of the dissolution vessel?
9. Section 2.4: What type of apparatus was used?
10. Section 2.4: How was material added to the media?
11. Section 2.4: What was the dose of material added? Also provide correlation coefficient information for calibration curve and concentration range.
12. Section 2.4: Please provide the equations for release kinetics that were calculated.
13. Please include a detailed methodology for molecular docking.
14. The methodologies for various studies were not discussed in the manuscript. Please provide the detailed methodology for FTIR, DSC, XRD, SEM analysis and for any additional studies for which the methodologies were not provided.
15. Authors are suggested to move any methodologies from the supplementary document into the main content.
Author Response
Reviewer#1
The current research article focusing on development of cyclodextrin inclusion complexes of nitazoxanide for improving oral bioavailability and anti-viral activity is interesting and matches the scope of the journal. However, the current version of the manuscript cannot be accepted for publication. The manuscript needs to be revised in various sections of the methodologies for better understanding. The authors are suggested to address the below comments:
We thank Reviewer#1 for the favorable reception of our work and for highlighting the positive points in our study. We have revised our manuscript taking into great consideration all the comments and suggestions. Thank you for helping us to improve our manuscript.
Comment 1
What is the BCS class of the drug?
Response 1
Thank you very much for your valuable and insightful suggestions. As suggested, we have provided the detailed explanations about the BCS class of the drug. Please refer to the details below, or see the changes highlighted in the revised manuscript.
Nitazoxanide (NTX) is classified as a Biopharmaceutics Classification System (BCS) Class IV drug. This means it exhibits low solubility and low permeability. NTX is poorly soluble in aqueous environments, which significantly limits its dissolution and bioavailability. This is a major challenge for its formulation into effective dosage forms. Despite its antiviral and antiparasitic efficacy, NTX has limited permeability across biological membranes, which can reduce its absorption and systemic availability when administered orally. This classification emphasizes the need for advanced formulation strategies, like inclusion complexes with β-cyclodextrins (β-CD and Hβ-CD) to enhance its solubility and permeability, thereby improving its therapeutic effectiveness.
Comment 2
Line 36: Be specific which type of cyclodextrin complex is effective.
Response 2
Thank you very much for your valuable and insightful suggestions. As suggested, we have mentioned the effective cyclodextrin complex in abstract section. Please refer to the details below for your reference only.
These findings suggest that Hβ-CD:NTX inclusion complexes may serve as effective carriers for delivering NTX in HSV-1 treatments using Vero cell models.
Comment 3
Authors have focused much on discussing the poor solubility of Nitazoxanide but have not provided any information such as BCS class, aqueous solubility, pKa, log P, any marketed oral formulations and doses available.
Response 3
Thank you very much for your valuable and insightful suggestions. As suggested, we have provide the detailed discussion about the poor solubility of NTX. Please refer to the details below, or see the changes highlighted in the revised manuscript.
The authors likely focused extensively on discussing the poor solubility of nitazoxanide (NTX) because it is the primary challenge limiting its bioavailability and clinical effectiveness, which the study aimed to address using β-cyclodextrin (β-CD) inclusion complexes.
Nitazoxanide (NTX) has emerged as a promising candidate in this context because of its unique mechanism of action and broad-spectrum activity against various pathogens [6]. NTX is classified as a Biopharmaceutics Classification System (BCS) Class IV drug. This means it exhibits low solubility and low permeability. NTX is poorly soluble in aqueous environments, which significantly limits its dissolution and bioavailability. NTX was originally developed to treat gastrointestinal infections caused by protozoa but has demonstrated efficacy against several viral infections, making it an attractive option for further research for antiviral therapy [7]. Its mechanism of action involves the inhibition of pyruvate-ferredoxin oxidoreductase, which is crucial for the energy metabolism of various pathogens, including viruses [8]. The potential of NTX to act against multiple viral targets provides a dual advantage where it effectively inhibits viral replication and may also reduce the likelihood of resistance development, which is an issue with existing antiviral drugs. Furthermore, the ability to explore NTX in combination with established antiviral agents may provide synergistic effects that enhance therapeutic outcomes. Although NTX has considerable therapeutic potential, its clinical application is hindered by its poor water solubility, which substantially limits its bioavailability and thus reduces its therapeutic effectiveness [9]. The crystalline structure and hydrophobic nature of NTX contribute to its low solubility, leading to poor dissolution rates upon oral administration. This presents a critical barrier to achieving the adequate plasma concentration required for effective treatment. As a result, enhancing the solubility of NTX has become a focal point of pharmaceutical research. Various formulation strategies have been explored to improve the solubility and bioavailability of NTX [10]. Recent advances in drug delivery systems, including solid dispersions, lipid-based formulations, and nanotechnology, have shown promise in this regard [11]. Techniques such as spray-drying, freeze-drying, and the incorporation of surfactants have been investigated to enhance dissolution rates, thereby improving its absorption in the gastrointestinal tract [12]. NTX, as the active ingredient, is prepared in a solid form in both suspension and tablet formulations. However, the omission of critical physicochemical properties, such as BCS class (Class IV: low solubility, low permeability), aqueous solubility (reported as very low, approximately 1.65 mg/L in water), pKa (approximately 5.3, affecting its ionization and solubility), and log P (3.9, indicating moderate lipophilicity), leaves gaps in understanding the complete solubility challenges of NTX [13]. Additionally, without details on marketed oral formulations and dosing (commonly available as a 500 mg tablet for parasitic infections), readers lack context for comparing the current findings to existing formulations. Such information would provide a comprehensive baseline for assessing the impact of the cyclodextrin complexes on NTX solubility and therapeutic potential, and its absence limits the scope of the discussion and practical relevance of the findings. Therefore, enhancing the solubility of NTX could provide better therapeutic options for patients.
Comment 4
Do authors think co-evaporation is the appropriate term to describe the process of preparing drug and cyclodextrin complexes as described in section 2.2
Response 4
Thank you very much for your valuable and insightful suggestions. As suggested, we have corrected the aforementioned issues in this manuscript. Please refer to the details below, or see the changes highlighted in the revised manuscript.
2.2. Preparation of β-CDs:NTX Inclusion Complex
To prepare the β-CDs (β-CD and Hβ-CD) and NTX drug inclusion complex (β-CDs:NTX), the co-precipitation method was used, and a 1:1 molar ratio of β-CDs to NTX was established by dissolving 0.307 g of NTX in 20 mL of ethanol and 1 g of β-CDs in 30 mL of water. Both solutions were combined in a sealed glass vial and mixed with a magnetic stirrer, which was stirred at a moderate speed for 30 min at room temperature. Subsequently, the mixture was stirred at a speed of 600 r/min for an additional 2 h to ensure complete interaction. Following this, the final solution was refrigerated at 4°C for 36 h to promote the formation of the β-CDs:NTX inclusion complex. After this incubation period, the precipitated β-CDs:NTX inclusion complex was obtained through filtration and thoroughly washed with ethanol to eliminate any uncomplexed NTX drug. The remaining residue was then vacuum-dried for 48 h, thus preparing the β-CDs:NTX inclusion complex for further studies.
Comment 5
Section 2.3: what type of filter was used? Is it hydrophilic or hydrophobic?
Response 5
Thank you very much for your valuable and insightful suggestions. As suggested, we have corrected the aforementioned sentences in the introduction section. Please refer to the details below for your reference only.
Given that the solvent used in this phase solubility study is aqueous, we utilized a hydrophilic membrane filter (Fisherbrand™ Membrane Filter, 0.45 μm) to ensure efficient filtration and compatibility with the solvent system.
Comment 6
Section 2.3: At what wavelength were the samples measured? Also please provide additional information pertaining to the calibration curve.
Response 6
Thank you very much for your valuable and insightful suggestions. As suggested, we have provided all the aforementioned issues in this phase solubility studies. Please refer to the details below, or see the changes highlighted in the revised manuscript.
2.5. Phase Solubility Studies
To perform phase solubility studies of β-CD and its derivative Hβ-CD with NTX, the procedures previously described by Higuchi and Connors [28] were followed. In this approach, 2 g of NTX is added to 25 mL of aqueous solutions containing various concentrations of β-CD or Hβ-CD (0, 0.002, 0.004, 0.006, 0.008, 0.010, and 0.012 mol/L) in a flask. The mixture was agitated at 30°C for three days to reach dissolution equilibrium. After this period, the solution was filtered through a 0.45-μm hydrophilic membrane (Fisherbrand™ Membrane Filter) filter to remove any undissolved NTX. The concentration of the dissolved NTX in each filtrate was measured at 280 nm via spectrophotometry using a UV-3220 Optizen spectrophotometer. The phase-solubility profile was obtained by plotting the solubility of NTX versus the concentration of β-CD and Hβ-CD. The apparent stability constant (KS) was determined from the slope of the linear phase-solubility diagrams.
Comment 7
Section 2.3: Cite the reference got Higuchi and Connors method.
Response 7
Thank you very much for your valuable and insightful suggestions. As suggested, we have cited the reference for the phase solubility studies conducted following the Higuchi and Connors method. Please refer to the details below, or see the changes highlighted in the revised manuscript.
To perform phase solubility studies of β-CD and its derivative Hβ-CD with NTX, the procedures previously described by Higuchi and Connors [28] were followed.
-
- Higuchi, T. K.; and A. Connors. Phase-solubility techniques. 1965, 4, 212–217.
Comment 8
Section 2.4: What’s the volume of the dissolution vessel?
Response 8
Thank you very much for your helpful comments. In response, we have included the details regarding the volume of the dissolution vessel in the manuscript as requested. Please refer to the details below, or see the changes highlighted in the revised manuscript.
A precisely weighed sample of the β-CD:NTX and Hβ-CD:NTX inclusion complex containing the equivalent of 25 mg NTX was placed in a 100 mL dissolution vessel containing with 20 mL of phosphate-buffered saline (PBS) at pH 7.4 and maintained at 37°C.
Comment 9
Section 2.4: What type of apparatus was used?
Response 9
Thank you very much for your comments. In response, we have included the details of the apparatus used in this study. Please refer to the details below for your reference only.
The apparatus used for the in vitro drug release study is a dissolution apparatus (708-DS). Specifically, it appears to be a rotating apparatus, commonly used in drug release testing, where the dissolution vessel is set to rotate at a constant speed of 100 rpm. This setup is designed to simulate the mixing environment of bodily fluids (such as in the gastrointestinal tract), ensuring proper agitation of the inclusion complex in the dissolution medium (PBS at pH 7.4). This type of apparatus is typically part of a USP dissolution tester (United States Pharmacopeia), which is often used in pharmaceutical studies to assess drug release kinetics.
Comment 10
Section 2.4: How was material added to the media?
Response 10
Thank you very much for your comments. In response, we have provided the details of the material added to the media used in this study. Please refer to the details below for your reference only.
The material (β-CD:NTX and Hβ-CD:NTX inclusion complexes) was added to the dissolution media by first precisely weighing the inclusion complex. This weighed sample was then placed into a 100 mL dissolution vessel containing 20 mL of phosphate-buffered saline (PBS) at pH 7.4, which mimics physiological conditions. The dissolution vessel was maintained at 37°C to simulate human body temperature, and the dissolution medium was agitated using a rotating apparatus set to 100 rpm to replicate the gentle stirring environment of bodily fluids. Over predetermined time intervals, 1.0 mL samples were withdrawn from the dissolution vessel, and each sample was replaced with an equal volume of fresh PBS to maintain the total volume in the vessel. This procedure allowed for continuous monitoring of the drug release over time, providing valuable data on the controlled release of NTX from the inclusion complexes
Comment 11
Section 2.4: What was the dose of material added? Also provide correlation coefficient information for calibration curve and concentration range.
Response 11
Thank you very much for your comments. In response, we have provided the details of all the aforementioned in this study. Please refer to the details below for your reference only.
The sample of β-CD:NTX and Hβ-CD:NTX inclusion complexes each contained the equivalent of 25 mg of NTX. This is the dose of the material added to the dissolution vessel. The correlation coefficient for the calibration curve, which quantifies the linearity of the relationship between the concentration of NTX and its absorbance at 422 nm.
Comment 12
Section 2.4: Please provide the equations for release kinetics that were calculated.
Response 12
Thank you very much for your valuable and insightful suggestions. As suggested, we have corrected the aforementioned sentences in this study. Please refer to the details below, or see the changes highlighted in the revised manuscript.
2.6. In Vitro Drug Release
The in vitro drug release study of the β-CD:NTX and Hβ-CD:NTX inclusion complexes was performed to evaluate the controlled release of NTX under physiological conditions. A precisely weighed sample of the β-CD:NTX and Hβ-CD:NTX inclusion complex containing the equivalent of 25 mg NTX was placed in a 100 mL dissolution vessel with 20 mL of phosphate-buffered saline (PBS) at pH 7.4 and maintained at 37°C. The dissolution apparatus (708-DS) was set to rotate at a constant speed of 100 rpm to ensure proper mixing and to replicate the gentle stirring environment of bodily fluids. At predetermined time intervals (0.5, 2, 5, 10, 15, 20, 25, 30, 35, 40, 45, 50, 60, 70, 80, and 90 min), 1.0 mL samples of the dissolution medium were carefully withdrawn using a syringe and immediately replaced with an equal volume of fresh PBS to maintain the total volume and prevent fluctuation in the concentration. The withdrawn samples were filtered through a 0.45-µm hydrophilic membrane filter to remove any undissolved particles. The concentration of NTX in the filtered samples was measured using UV-visible spectrophotometry at a wavelength of 422 nm, which is specific for the detection of NTX. The cumulative amount of drug released over time was calculated using a calibration curve prepared from known concentrations of NTX. The release profile of NTX from the β-CD:NTX and Hβ-CD:NTX inclusion complexes was plotted as the cumulative percentage of drug release rate calculated using Equation (2).
Comment 13
Please include a detailed methodology for molecular docking.
Response 13
Thank you very much for your valuable and insightful suggestions. As suggested, we have provided the detailed methodology of the molecular docking studies. Please refer to the details below, or see the changes highlighted in the revised manuscript.
2.4. Molecular docking
To investigate the binding interactions between NTX and β-Cyclodextrins (β-CDs), docking simulations were performed with the help of PatchDock and FireDock servers. Initially, three-dimensional models of NTX and β-CDs (including both β-CD and Hβ-CD) were constructed using Chem3D 21.0.0 software. These models were then saved in PDB format for compatibility with the docking software. Prior to docking, energy minimization was conducted on both the ligand (β-CDs) and the receptor (NTX) to reduce steric clashes and optimize the molecular geometry. The optimized structures were uploaded to the PatchDock server (http://bioinfo3d.cs.tau.ac.il/PatchDock/), which uses a shape complementarity algorithm to predict potential interactions based on surface compatibility. The PatchDock-generated β-CD complexes were subsequently refined on the FireDock server (http://bioinfo3d.cs.tau.ac.il/FireDock/), which evaluates complex stability and binding affinity through an energy-based scoring mechanism. Finally, the top-scoring docked configurations were chosen for further analysis in UCSF Chimera 1.8.1 (https://www.cgl.ucsf.edu/chimera), enabling a detailed examination of structural interactions.
Comment 14
The methodologies for various studies were not discussed in the manuscript. Please provide the detailed methodology for FTIR, DSC, XRD, SEM analysis and for any additional studies for which the methodologies were not provided.
Response 14
Thank you very much for your valuable and insightful suggestions. As suggested, we have provided the detailed methodology for FTIR, DSC, XRD, SEM analysis. Please refer to the details below, or see the changes highlighted in the revised manuscript.
2.3. Characterization Techniques
UV-Vis absorption spectra were measured using an Optizen UV 3220 spectrometer. Solution samples were scanned from 200 to 550 nm at a speed of 240 nm/min, using a quartz cell with a path length of 1.0 cm. FTIR analysis was conducted on β-CD, Hβ-CD, NTX, and their respective inclusion complexes (β-CD:NTX and Hβ-CD:NTX) using KBr pellets and a System 2000 FTIR instrument from Perkin Elmer, covering a spectral range from 4000 to 400 cm⁻¹. X-ray diffraction (XRD) patterns of β-CD, Hβ-CD, NTX, and their inclusion complexes were recorded using an X-ray diffractometer with CuKα radiation. The instrument settings included a voltage of 30 mA, a current of 30 kV, and a scan rate of 2° min⁻¹ within a 2θ angle range of 10-80°. The morphology of β-CD, Hβ-CD, NTX, and their inclusion complexes was examined via scanning electron microscopy (SEM) with a LEO 1430 microscope from Zeiss (Germany). Samples were coated with gold using an Emitech K 550X sputter coater to improve conductivity before SEM analysis. Thermogravimetric analysis (TGA) was conducted to assess the thermal stability of β-CD, Hβ-CD, NTX, and their inclusion complexes. This analysis was performed with a Shimadzu thermogravimetric analyzer and a Perkin Elmer DSC 7 system. Samples weighing approximately 5–6 mg were heated in alumina crucibles from 50 to 400 °C at a rate of 10 °C/min under a nitrogen atmosphere.
Comment 15
Authors are suggested to move any methodologies from the supplementary document into the main content.
Response 15
Thank you very much for your valuable and insightful suggestions. As suggested, we have moved the methodologies from the supplementary document into the main manuscript. Please refer to the details below, or see the changes highlighted in the revised manuscript.
2.3. Characterization Techniques
UV-Vis absorption spectra were measured using an Optizen UV 3220 spectrometer. Solution samples were scanned from 200 to 550 nm at a speed of 240 nm/min, using a quartz cell with a path length of 1.0 cm. FTIR analysis was conducted on β-CD, Hβ-CD, NTX, and their respective inclusion complexes (β-CD:NTX and Hβ-CD:NTX) using KBr pellets and a System 2000 FTIR instrument from Perkin Elmer, covering a spectral range from 4000 to 400 cm⁻¹. X-ray diffraction (XRD) patterns of β-CD, Hβ-CD, NTX, and their inclusion complexes were recorded using an X-ray diffractometer with CuKα radiation. The instrument settings included a voltage of 30 mA, a current of 30 kV, and a scan rate of 2° min⁻¹ within a 2θ angle range of 10-80°. The morphology of β-CD, Hβ-CD, NTX, and their inclusion complexes was examined via scanning electron microscopy (SEM) with a LEO 1430 microscope from Zeiss (Germany). Samples were coated with gold using an Emitech K 550X sputter coater to improve conductivity before SEM analysis. Thermogravimetric analysis (TGA) was conducted to assess the thermal stability of β-CD, Hβ-CD, NTX, and their inclusion complexes. This analysis was performed with a Shimadzu thermogravimetric analyzer and a Perkin Elmer DSC 7 system. Samples weighing approximately 5–6 mg were heated in alumina crucibles from 50 to 400 °C at a rate of 10 °C/min under a nitrogen atmosphere.
2.4. Molecular docking
To investigate the binding interactions between NTX and β-Cyclodextrins (β-CDs), docking simulations were performed with the help of PatchDock and FireDock servers. Initially, three-dimensional models of NTX and β-CDs (including both β-CD and Hβ-CD) were constructed using Chem3D 21.0.0 software. These models were then saved in PDB format for compatibility with the docking software. Prior to docking, energy minimization was conducted on both the ligand (β-CDs) and the receptor (NTX) to reduce steric clashes and optimize the molecular geometry. The optimized structures were uploaded to the PatchDock server (http://bioinfo3d.cs.tau.ac.il/PatchDock/), which uses a shape complementarity algorithm to predict potential interactions based on surface compatibility. The PatchDock-generated β-CD complexes were subsequently refined on the FireDock server (http://bioinfo3d.cs.tau.ac.il/FireDock/), which evaluates complex stability and binding affinity through an energy-based scoring mechanism. Finally, the top-scoring docked configurations were chosen for further analysis in UCSF Chimera 1.8.1 (https://www.cgl.ucsf.edu/chimera), enabling a detailed examination of structural interactions.

Reviewer 2 Report
Comments and Suggestions for Authors
The manuscript presents a comprehensive study on the formulation and characterization of β-cyclodextrin and hydroxypropyl-β-cyclodextrin with nitazoxanide. The authors aim to improve the solubility and bioavailability of nitazoxanide and employ different techniques, including UV-Vis spectroscopy, FTIR, XRD, SEM, TGA, and DSC, to validate the formation of inclusion complexes and assess their stability and release profiles. Please, provide the method for docking study as well. While the manuscript is generally well-structured, some figures and tables lack sufficient explanation in the context. Ensure that all figures and tables are adequately referenced and explained in the text. Please, provide the potential mechanisms by which β-CD and Hβ-CD enhance nitazoxanide’s solubility and bioavailability and compare the findings with other methods.
Author Response
Reviewer#2
The manuscript presents a comprehensive study on the formulation and characterization of β-cyclodextrin and hydroxypropyl-β-cyclodextrin with nitazoxanide. The authors aim to improve the solubility and bioavailability of nitazoxanide and employ different techniques, including UV-Vis spectroscopy, FTIR, XRD, SEM, TGA, and DSC, to validate the formation of inclusion complexes and assess their stability and release profiles. Please, provide the method for docking study as well. While the manuscript is generally well-structured, some figures and tables lack sufficient explanation in the context. Ensure that all figures and tables are adequately referenced and explained in the text. Please, provide the potential mechanisms by which β-CD and Hβ-CD enhance nitazoxanide’s solubility and bioavailability and compare the findings with other methods.
We thank Reviewer#2 for the favorable reception of our work and for highlighting the positive points in our study. We have revised our manuscript taking into great consideration all the comments and suggestions. Thank you for helping us to improve our manuscript.
Comment 1
Provide the method for docking study as well.
Response 1
Thank you very much for your valuable and insightful suggestions. As suggested, we have provided the detailed methodology of the molecular docking studies. Please refer to the details below, or see the changes highlighted in the revised manuscript.
2.4. Molecular docking
To investigate the binding interactions between NTX and β-Cyclodextrins (β-CDs), docking simulations were performed with the help of PatchDock and FireDock servers. Initially, three-dimensional models of NTX and β-CDs (including both β-CD and Hβ-CD) were constructed using Chem3D 21.0.0 software. These models were then saved in PDB format for compatibility with the docking software. Prior to docking, energy minimization was conducted on both the ligand (β-CDs) and the receptor (NTX) to reduce steric clashes and optimize the molecular geometry. The optimized structures were uploaded to the PatchDock server (http://bioinfo3d.cs.tau.ac.il/PatchDock/), which uses a shape complementarity algorithm to predict potential interactions based on surface compatibility. The PatchDock-generated β-CD complexes were subsequently refined on the FireDock server (http://bioinfo3d.cs.tau.ac.il/FireDock/), which evaluates complex stability and binding affinity through an energy-based scoring mechanism. Finally, the top-scoring docked configurations were chosen for further analysis in UCSF Chimera 1.8.1 (https://www.cgl.ucsf.edu/chimera), enabling a detailed examination of structural interactions.
Comment 2
While the manuscript is generally well-structured, some figures and tables lack sufficient explanation in the context. Ensure that all figures and tables are adequately referenced and explained in the text.
Response 2
Thank you very much for your valuable and insightful suggestions. Following your recommendations, we have addressed all the issues mentioned and made the necessary corrections throughout the manuscript. We believe these revisions have enhanced the clarity and rigor of our work. Please see the changes highlighted in the revised manuscript.
Comment 3
Provide the potential mechanisms by which β-CD and Hβ-CD enhance nitazoxanide’s (NTX) solubility and bioavailability and compare the findings with other methods.
Response 3
Thank you very much for your valuable and insightful suggestions. As suggested, we have provide the detailed potential mechanisms of β-CD and Hβ-CD enhance NTX solubility and bioavailability and compare the findings with other methods. Please refer to the details below, or see the changes highlighted in the revised manuscript. We believe these revisions have enhanced the clarity and rigor of our work.
3.11. Mechanism
The enhancement of NTX solubility and bioavailability by β-CD and Hβ-CD is achieved primarily through the formation of inclusion complexes, where NTX is encapsulated within the β-CDs hydrophobic cavity. This encapsulation not only shields NTX from aqueous environments, reducing aggregation and crystallization, but also improves its dispersion, creating an amorphous state with increased surface area for dissolution [47]. The β-CDs act as molecular containers, stabilizing NTX and enhancing its solubility without requiring harsh solvents or additional stabilizers [16]. Hβ-CD, with its modified hydrophilic structure, enhances NTX dissolution further by facilitating faster and more complete release, which can increase NTX drug concentration at absorption sites, improving membrane permeability and cellular uptake [48–51]. Compared with other solubility enhancement methods like solid dispersions [52], nanoparticles [53,54], and liposomal systems [55], β-CDs offer advantages in simplicity, safety, and stability, especially under physiological conditions [56,57]. Moreover, β-CD and Hβ-CD complexes provide sustained NTX release and improved bioavailability without the physical instability or toxicity risks associated with co-solvents, pH modification, or surfactants. These properties make β-CDs particularly effective for NTX delivery, as they enhance solubility, stability, and bioavailability in a balanced, biocompatible system suitable for antiviral therapies.
References
- Schoeman, C.; van Niekerk, S.; Liebenberg, W.; Hamman, J. Cyclodextrin Inclusion Complex and Amorphous Solid Dispersions as Formulation Approaches for Enhancement of Curcumin’s Solubility and Nasal Epithelial Membrane Permeation. Future Journal of Pharmaceutical Sciences 2024, 10, doi:10.1186/s43094-024-00656-8.
- Prado, A.R.; Yokaichiya, F.; Franco, M.K.K.D.; Silva, C.M.G. da; Oliveira-Nascimento, L.; Franz-Montan, M.; Volpato, M.C.; Cabeça, L.F.; de Paula, E. Complexation of Oxethazaine with 2-Hydroxypropyl-β-Cyclodextrin: Increased Drug Solubility, Decreased Cytotoxicity and Analgesia at Inflamed Tissues. Journal of Pharmacy and Pharmacology 2017, 69, 652–662, doi:10.1111/jphp.12703.
- Psimadas, D.; Georgoulias, P.; Valotassiou, V.; Loudos, G. Molecular Nanomedicine Towards Cancer : Journal of pharmaceutical sciences 2012, 101, 2271–2280, doi:10.1002/jps.
- Lima, B. dos S.; Campos, C. de A.; da Silva Santos, A.C.R.; Santos, V.C.N.; Trindade, G. das G.G.; Shanmugam, S.; Pereira, E.W.M.; Marreto, R.N.; Duarte, M.C.; Almeida, J.R.G. da S.; et al. Development of Morin/Hydroxypropyl-β-Cyclodextrin Inclusion Complex: Enhancement of Bioavailability, Antihyperalgesic and Anti-Inflammatory Effects; Elsevier Ltd, 2019; Vol. 126; ISBN 5579991024838.
- Chauhan, R.; Madan, J.; Kaushik, D.; Sardana, S.; Pandey, R.S.; Sharma, R. Inclusion Complex of Colchicine in Hydroxypropyl-β-Cyclodextrin Tenders Better Solubility and Improved Pharmacokinetics. Pharmaceutical Development and Technology 2013, 18, 313–322, doi:10.3109/10837450.2011.591801.
- Singh, A.; Worku, Z.A.; Van Den Mooter, G. Oral Formulation Strategies to Improve Solubility of Poorly Water-Soluble Drugs. Expert Opinion on Drug Delivery 2011, 8, 1361–1378, doi:10.1517/17425247.2011.606808.
- Krosuri, P.K. Recent Advances In The Development Of Nanoparticles In Enhancement Of Solubility Of Poorly Soluble Drugs. Journal of Pharmaceutical Negative Results 2022, 13, 3512–3527, doi:10.47750/pnr.2022.13.S10.422.
- Liu, Y.; Liang, Y.; Yuhong, J.; Xin, P.; Han, J.L.; Zhu, R.; Zhang, M.; Chen, W.; Ma, Y.; Du, Y.; et al. Advances in Nanotechnology for Enhancing the Solubility and Bioavailability of Poorly Soluble Drugs. Drug Design, Development and Therapy 2024, 18, 1469–1495, doi:10.2147/DDDT.S447496.
- Lee, M.K. Liposomes for Enhanced Bioavailability of Water-Insoluble Drugs: In Vivo Evidence and Recent Approaches. Pharmaceutics 2020, 12, doi:10.3390/pharmaceutics12030264.
- Charumanee, S.; Okonogi, S.; Sirithunyalug, J.; Wolschann, P.; Viernstein, H. Effect of Cyclodextrin Types and Co-Solvent on Solubility of a Poorly Water Soluble Drug. Scientia Pharmaceutica 2016, 84, 694–704, doi:10.3390/scipharm84040694.
- de Miranda, J.C.; Martins, T.E.A.; Veiga, F.; Ferraz, H.G. Cyclodextrins and Ternary Complexes: Technology to Improve Solubility of Poorly Soluble Drugs. Brazilian Journal of Pharmaceutical Sciences 2011, 47, 665–681, doi:10.1590/S1984-82502011000400003.

Round 2
Reviewer 1 Report
Comments and Suggestions for Authors
All the comments are well addressed with supporting literature and justification. The revised version can be accepted for publication.